# A Rare Case of Cervical Spinal Arteriovenous Malformation: A Case Report

**DOI:** 10.3390/medicina60061011

**Published:** 2024-06-20

**Authors:** Jolanta Ošiņa, Kristaps Jurjāns, Kārlis Kupčs, Tatjana Rzajeva, Evija Miglāne

**Affiliations:** 1Faculty of Residency, Riga Stradins University, LV-1007 Riga, Latvia; 2Neurology Department, Pauls Stradins Clinical University Hospital, LV-1002 Riga, Latvia; kristaps.jurjans@stradini.lv (K.J.); tatjana.rzajeva@stradini.lv (T.R.); evija.miglane@stradini.lv (E.M.); 3Department of Neurology and Neurosurgery, Riga Stradins University, LV-1007 Riga, Latvia; 4Institute of Diagnostic Radiology, Pauls Stradins Clinical University Hospital, LV-1002 Riga, Latvia; karlis.kupcs@stradini.lv

**Keywords:** spinal arteriovenous malformation, neurovascular disease, cervical embolization, arteriovenous malformation surgery

## Abstract

Arteriovenous malformation (AVM) is an abnormal connection of vasculature resulting in capillary bed bypassing and leading to neurological deterioration and high risk of bleeding. Intramedullary AVMs in the cervical spinal cord are rare and require precise diagnostics and treatment. We present a clinical case of recurrent AVMs in a 28-year-old Caucasian female with sudden and severe neck pain and variable neurological symptoms along with current diagnostic and treatment modalities. Conservative treatment was partially effective. MRI and DSA confirmed AVMs at C4 level with subsequent several endovascular treatment sessions at the age of 15 and 24 with mild neurological improvement. Afterwards the patient underwent rehabilitation with minor neurological improvement. This case highlights the clinical progression and treatment of AVMs along with showcasing current pathophysiology, classification, and imaging.

## 1. Introduction

Spinal arteriovenous malformations (AVMs) are developmental or acquired abnormal direct connections of normal-sized to enlarged radiculomedullary arteries with enlarged tortuous radiculomedullary veins, without an intervening capillary network [1]. One mutated gene (MAP2K1), which causes hereditary hemorrhagic telangiectasia (HHT)1, has been identified to be responsible for inherited forms of extracranial arteriovenous malformation [2].

Spinal AVMs are rare but have potentially devastating neurological outcomes if not diagnosed and treated accordingly. AVMs usually occur in the mid-20s, but about 20% of the lesions are diagnosed before the age of 16. Data on incidence are scarce and vary approximately between 0.2/100,000 and 0.3/100,000 [3,4,5,6]. The most common variety among spinal vascular malformations is dural arteriovenous fistula, which accounts for 70 to 80% of them [1,6,7].

## 2. Pathophysiology

Intramedullary spinal AVMs are supplied by medullary arteries (anterior and posterior spinal artery), drain through medullary veins, and are characterized by high pressure, relatively low resistance, and high blood flow. Ultimately this leads to venous hypertension, which, in turn, can precipitate many neurological deficits secondary to mass effect and normal spinal blood flow disruption. Associated arterial aneurysms are common but in rare cases can lead to hemorrhaging [1,3,6,8,9].

## 3. Classification

There are multiple AVMs classification schemes. According to the recent and commonly accepted classification system, AVMs are divided into three categories depending on characteristic, pathophysiology, clinical presentation, diagnostic modality, and previous nomenclature [1,3,7,10] Table 1.

## 4. Clinical Picture

The signs and symptoms of AVMs are attributed to mass effect and ischemia, and they may vary. Common symptoms include local, radicular, or diffuse pain depending on the segments of the spinal cord involved. Patients often present with sensory and/or motor disturbances. The onset of neurological symptoms may be progressive or abrupt, usually, in the case of hemorrhaging [1,6,11].

## 5. Imaging

In cases of suspected AVMs, the investigation method of choice is magnetic resonance imaging (MRI) with magnetic resonance angiography (MRA). It can assess the spinal cord and surrounding structures, thus helping to narrow the differential diagnosis. The common signs in MRI of AVMs are the presence of spinal cord oedema with increased T2 signal, flow voids, and dilated intervertebral veins in T2—weighted imaging. MRA is usually supplemental to identify the number of possible arterial feeders that supply the AVMs. Afterwards, the definitive radiological procedure in the pretreatment evaluation of AVMs is selective spinal catheter digital subtraction angiography (DSA) [1,6,9,10,11,12].

## 6. Treatment

Treatment is complex and largely dependent on the angioarchitecture of the AVMs, as well as prior treatment approaches and its success rate of reversing AVMs neurological morbidity. Usually, the treatment of AVMs is conducted via endovascular embolization and/or surgery [1,3,6,11,13]. Endovascular embolization and surgery success rate of complete obliteration of AVMs vary depending on the type. In intramedullary AVMs, complete obliteration is seen in about 33 to 38% and recanalization of AVMs is a common problem requiring repeated embolization and/or surgery. The complete obliteration rate of spinal AVMs after conventional neurosurgery reaches 78%, but the complication rates compared to endovascular treatment are also significantly higher [14,15,16,17,18]. There have been trials to treat AVMs using stereotactic radiosurgery with promising results, indicating that it might be an alternative management strategy for patients who cannot be treated using an endovascular approach or neurosurgery [19,20].

It is difficult to determine the superior treatment modality, but the current approach is to combine endovascular embolization prior neurosurgery with complete or partial resection for better neurological outcome and obliteration rate [1,3,6,10,11,13,15,16,17,18,19,20,21].

## 7. Clinical Case

### 7.1. First Episode (2011)

A fifteen-year-old Caucasian female, during school lessons, experienced a sudden onset of severe pain in the neck. The pain was irradiating to the left arm and to the right upper arm. There was no evidence of trauma or any other diseases during the last two weeks. The patient was admitted to the Children Clinical University Hospital, Riga, Latvia.

On the initial neurological examination, she was conscious and fully oriented. There was palpatory pain in both trapezius muscles (left > right). Muscle strength in left arm was grade 4 and grade 5 in the right arm. The range of motion was diminished in both arms due to severe pain. Hypoesthesia was seen at the C4 to C5 dermatome level. There was tendon reflex asymmetry in both arms (left > right). The patient had no family history of AVMs or any other hereditary blood vessel anomalies.

A spinal cord MRI at the neck and thoracolumbar level was performed. MRI showed diffuse spinal cord oedema from C2 to Th4 with differential diagnosis of AVMs or hemangioblastoma (Figure 1). DSA was performed as a next step and diagnosis of intramedullary arteriovenous malformation at the 4th cervical segmentary level was conducted (Figure 2A). The patient received peroral and intravenous analgesics for pain management. During the patient’s time at the hospital, neurological deficits increased up to left-side hypoesthesia, autonomic dysfunction, and mild paresis in the right arm.

With recommendations for pain treatment and exemption from physical activities, the patient was discharged from hospital. An ambulatory interventional radiologist consultation and a spinal arteriovenous malformation embolization were scheduled.

After two months from the first symptoms, spinal AVM embolization with Onyx™ liquid embolic system was successfully performed (Figure 2B; Figure 3). After five months, a control DSA was performed, and results showed a small residual AVM part with a leading branch from the left vertebral artery. One year later, another MRI and DSA were performed showing a positive dynamic—the decrement of residual AVM (Figure 4).

### 7.2. Second Episode (2020)

Nine years later the patient was admitted to P. Stradins Clinical University Hospital with severe, pulling pain in neck region, pain in right upper arm, left hemi-hypoesthesia, and paresthesia.

Neurological examination showed hypoesthesia in the left leg and left palm and palpatory tenderness in the neck muscles. Tendon reflexes were asymmetric (right > left). There was also a positive Babinski sign on the right.

Spinal cord MRI for the neck region showed intramedullary oedema from C3 to C6. DSA showed residual AVM nidus enlargement compared to the previous investigations (Figure 5A). Based on evidence of progressing neurological symptoms, severe pain syndrome, and radiological findings, a decision was made to conduct repeated AVM embolization using a “Phil” embolization system, which led to complete AVM occlusion (Figure 5B).

After the procedure, the patient still complained about severe pain in the neck and shoulder muscles. Neurological examination revealed asymmetrical tetraparesis with severe weakness in the right distal arm, moderately severe weakness in the right leg, and mild weakness in the left proximal arm. There was hypoesthesia in the right side of the body.

Treatment was continued with dehydrating and analgesic drugs and rehabilitation with a physiotherapist. The patient was discharged with clinical improvement. At the time of discharge, neurological examination indicated asymmetrical tetraparesis with mild paresis in the proximal left arm and leg, moderate paresis proximal and severe paresis in the distal right arm, and moderate paresis in the right leg with spasticity and clonus. The sensory examination showed mild superficial sensory deficits in the right arm and left leg. There was disturbed position sense in the right leg. Tendon reflexes in the arms were symmetrical, but in the legs they were increased in the right side and a positive Babinski sign was observed. She was able to sit up in the bed without assistance.

The patient was transferred to the rehabilitation center.

One year later a control MRI for the spinal cord neck region was made. The finding was similar, compared to previous MRI. There were no data about oedema. MRI visualized the embolization material located more in the central and right paramedian part of the spinal cord as well as atrophy of anterior column at the level C3–C4. A small, local residual AVM was suspected.

Control DSA revealed total AVM occlusion with an irrelevant arteriovenous shunt at the caudal part of malformation.

### 7.3. Third Episode (2023)

In 2023, during cooking, the patient felt a sudden tremor, feeling of heat all over the body, and aching pain in the neck region. The next day she felt a pulling pain all over the back and was seeking help in the emergency department. Neurological examination showed no deterioration compared to the previous data and no changes in blood tests. Therefore, there was no indication for admitting the patient to the neurological department.

After two days, the patient was admitted again in the emergency department, with complains of severe pain in the neck region that irradiated to the left arm and shooting pain along the whole spine. There also was increased numbness in left arm from third to fifth finger. The patient was admitted in the neurology department. Neurological examination showed no objective neurological deterioration. MRI and DSA of the cervical region of spinal cord were performed. MRI showed hypointense embolization material on right side at the C3–C4 level and gliosis with local scaring and atrophy around it as well as a mild oedema at the level of the C3 vertebral body (Figure 6). DSA showed insignificant enlargement of the residual AVM compared to DSA findings a year prior. The patient received dehydrating and analgesic drug treatment, and with clinical improvement after 12 days from the first symptoms, the patient was discharged and continued rehabilitation at a specialized rehabilitation center.

At the time of discharge, neurological examination showed mild (distal > proximal) right spastic paresis, hypoesthesia in the right arm, deep sensation disturbances in the left leg, and paresthesia in left side of the body. Tendon reflexes were asymmetrical, increased in the left arm and right leg. There was evidence of a positive Babinsky sign on the right side. The patient also complained about diminished sensation at the level of Th4-Th5 dermatomes.

## 8. Discussion

Spinal AVMs are a rare, heterogenous group of vascular malformations with a clinical presentation that depends on the affected level of the spinal cord and size of malformation. We present a case of a 28-year-old Caucasian female patient with a first episode at the age of 15, characterized by initial sudden neck pain irradiating to both proximal arms, unilateral sensory loss, and asymmetry of tendon reflexes. The neurological symptom presentation in our patient was similar to other case reports and literature reviews, according to the affected spinal cord level [1,8,10,12,17,21,22,23]. During the patient’s hospital stay, neurological symptoms progressed. MRI angiography and DSA revealed AVM at the C4 level. Since neurological symptoms in the AVMs are not specific and can mimic other neurological diseases with different treatment modalities and outcomes, MRI with or without angiography is the current imaging modality of choice in the evaluation of patients presenting with myelopathic symptoms in the search for spinal cord lesions [10,13]. When AVM diagnosis is confirmed, DSA is usually performed to evaluate involved blood vessels and to decide the definitive treatment option [9,10].

Two weeks later AVM embolization was performed, and after a year, control DSA showed decremental changes in AVM. Nine years later, she experienced another episode of neck pain and left hemisensory loss. MRI angiography and DSA revealed recanalization of previous AVM. Embolization of AVM was performed, which resulted in asymmetric tetraparesis and sensory loss in the right arm and left leg. At discharge from the hospital, there was a mild neurological improvement. In 2023. the patient was again hospitalized due to neck and back pain and asymmetric tetraparesis and hypoesthesia in right arm and leg, with no deterioration, as noted previously. MRI angiography and DSA showed residual AVMs with insignificant enlargement. Patients received dehydrating drugs and analgesics with clinical improvement and continued rehabilitation ambulatory. Our case highlights one of the endovascular treatment concerns—recanalization of AVMs. [10,15,24]. Endo T. et al. compared surgical and endovascular treatments. Their results were comparable in obliterating AVMs: 88% for surgical and 74% endovascular treatment. It is worth mentioning that even with modern use of newer embolization agent “Onxy”, complete obliteration of AVMs was seen in 37.5% [25]. It remains unclear whether endovascular treatment combined with surgical excision of the AVM lesions would give better results in this patient. On the one hand, endovascular embolization has fewer complications but higher AVMs recanalization rates. Surgery, on the other hand, if possible, offers excision of the AVMs lesion with much lower recanalization rates but higher risk of complications. M. Yashar et al. conducted a single-center retrospective study where four out of nine patients received combined endovascular and surgical treatment with 100% complete obliteration of the AVMs and neurological improvement [15]. Alim P. Mitha et al. also conducted a retrospective single-center study. Of 80 patients, 35% underwent laminectomy and 65% laminoplasty following midline myelotomy (twenty-four patients), lateral myelotomy (eight patients), or posterolateral myelotomy (thirty patients). Residual AVMs lesion was noted in 3 patients. Immediate improvement after surgery was seen in 6%. However, 11% of patients were worse and 83% remained at the same neurological status. A long-term follow-up was conducted with 62 patients, where 10% were worse, 68% remained at the same neurological status, and 26% had improvements [26].

Stereotactic radiosurgery has promising results in AVM treatment, but its role is yet to be established and more research is needed in this area [10,15,16,17,18,19,20,21]. Sherif Rashad et al. conducted a single-center observational study, performing radiosurgery with CyberKnife^TM^ in patients who were diagnosed with AVMs and who were not suitable for endovascular treatment or had only experienced partial effects. Four of five patients had neurological symptom improvement with no worsening during follow-up, as well as improvement of the spinal lesion in MRI and DSA [27]. A systematic review by Peter L. Zhan et al. did not confirm the superiority of stereotactic radiosurgery and addressed the issue of lacking prospective studies and the risk of irreversible radiation-induced myelitis. Although, it is worth mentioning that the success rate of neurological symptom improvement and AVM decrement was more than 92%, higher than in microsurgery and endovascular management [28].

Treatment of AVMs is complex and depends on many factors, such as the location and architecture of the lesion, previous treatment, and the experience of interventional radiologists and neurosurgeons. We avoided surgical treatment due to the high risk of neurological deterioration, and we decided to opt for endovascular treatment because of the excellent experience of our invasive radiologists.

## 9. Conclusions

We have presented a case of a 28-year-old Caucasian female patient, with three episodes of initial neck pain in total, from 2011 to 2023. The patient had initial sensory loss in half of their body with neurological deterioration afterwards up to asymmetric tetraparesis with bilateral sensory symptoms after the second embolization. MRI angiography initially demonstrated spinal cord oedema from C2 to Th4. Afterwards DSA confirmed AVMs at the level of C4 in the spinal cord, which were treated with several endovascular embolization sessions at the age of 15 and 24. This case highlights the clinical course and treatment of AVM lesions, along with showcasing of a classification system and current imaging modalities; it raises awareness of the potentially devastating neurological outcome if not recognized and treated promptly.

## Figures and Tables

**Figure 1 medicina-60-01011-f001:**
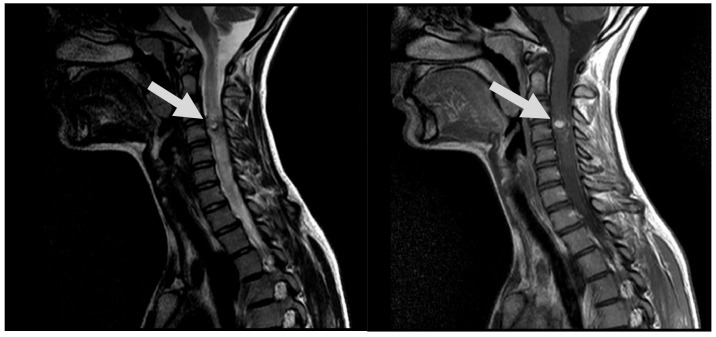
MRI, T2, and T1 CE sag demonstrates hypervascular contrast enhancing round-shaped lesion at C3 level with signs of hemorrhaging, as well as ascending and descending spinal cord oedema (arrows).

**Figure 2 medicina-60-01011-f002:**
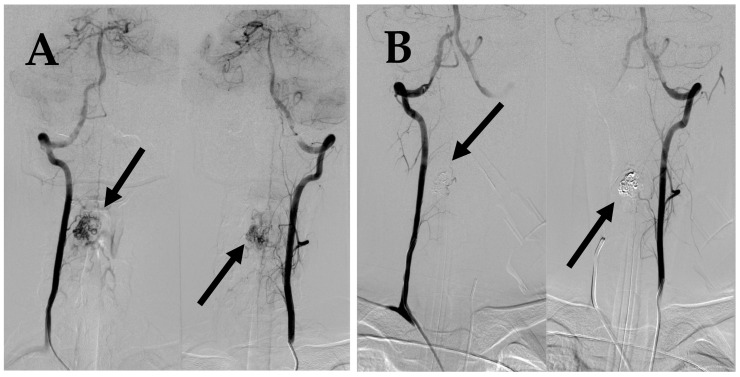
First DSA. (**A**) DSA before embolization demonstrates intramedullary AVM supplied by spinal branches from both vertebral arteries (arrows). (**B**) Post embolization DSA demonstrates complete occlusion of AVM (arrows).

**Figure 3 medicina-60-01011-f003:**
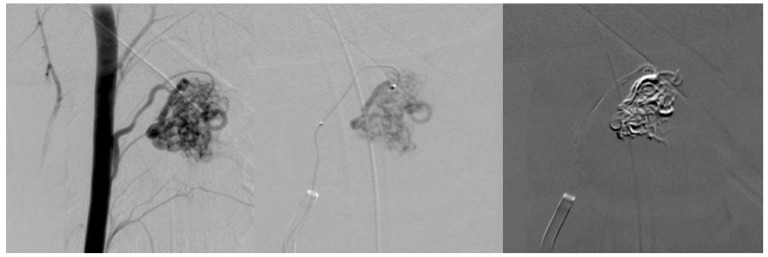
First embolization. Superselective catheterization of AVM nidus with microcatheter and subsequent embolization with ONYX liquid embolization system.

**Figure 4 medicina-60-01011-f004:**
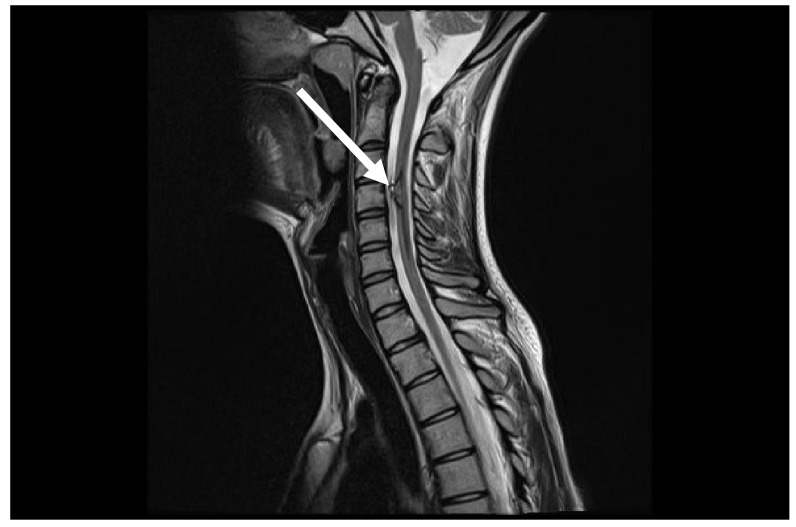
Control MRI after one year from first symptoms. MRI follow-up T2 sag demonstrates marked improvement with disappearance of spinal cord oedema (arrow).

**Figure 5 medicina-60-01011-f005:**
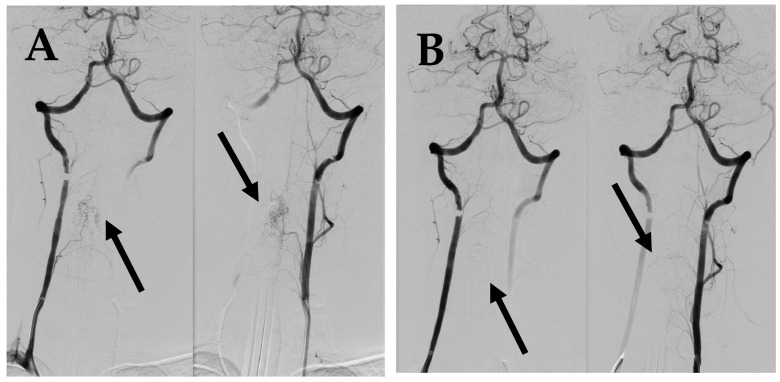
(**A**) DSA before 2nd embolization demonstrates intramedullary AVM supplied by spinal branches from both vertebral arteries (arrows). (**B**) Post embolization DSA demonstrates complete occlusion of AVM (arrows).

**Figure 6 medicina-60-01011-f006:**
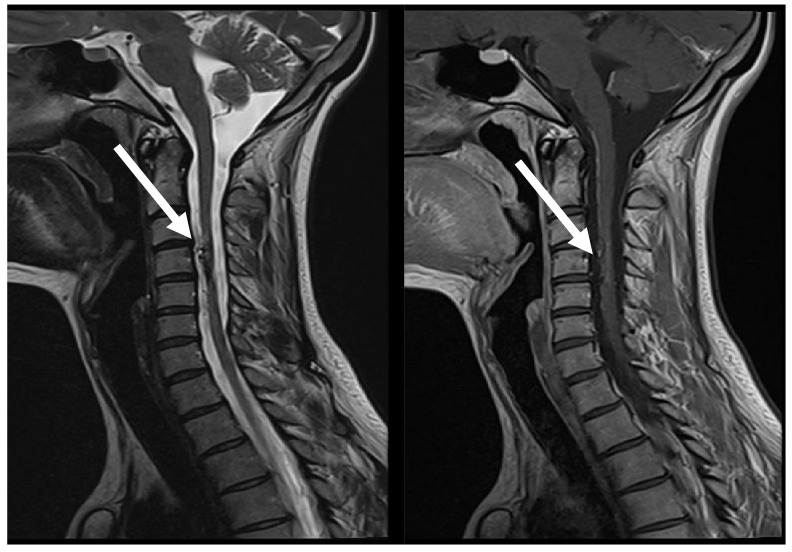
MRI follow-up T2 sag; T1 CE sag, demonstrate AVM at C3 post embolization, with evidence of segmental spinal cord atrophy (arrows).

**Table 1 medicina-60-01011-t001:** Proposed sAVM classification scheme by R.F. Spetzler et al.

Characteristic	Extradural–Intradural	Intramedullary	Conus Medullaris
Pathophysiology	Compression, vascular steal, hemorrhage	Compression, vascular steal, hemorrhage	Venous hypertension, compression, hemorrhage
Presentation	Pain, progressive myelopathy	Pain, acute myelopathy, progressive myelopathy	Progressive myelopathy, radiculopathy
Diagnostic modality	MR imaging, angiography, high–flow, multiple feeders	MR imaging, angiography	MR imaging, angiography
Previous nomenclature	Juvenile AVM, metameric AVM	Classic AVM, glomus type	None

## Data Availability

The original contributions presented in the study are included in the article, further inquiries can be directed to the corresponding author.

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
