# Peer review of "A Rare Case of Cervical Spinal Arteriovenous Malformation: A Case Report"

_medicina, 2024, doi:10.3390/medicina60061011_

Round 1
Reviewer 1 Report
Comments and Suggestions for Authors
General comments:
The manuscript titled “A Rare Case of Cervical Spinal Arteriovenous Malformation: A 2 Case Report” is well structured overall, and it was found that it adds to current knowledge, but it needs some modifications that we will point out in the comments sections below.
Overall, the relevance of the report was highlighted and no ethical violations were found. The title accurately described by including the phrase “a case report” in the title.
The abstract needs to be slightly improved, with a clearer background narrative before the case presentation and a conclusion at the end.
Major comments:
In the background,
1. Line 33, according to my knowledge AVMs occurs between arterioles and venules, without an intervening normal capillary network, but not between arteries and veins.
2. Line 38-39, you mentioned that AVMs usually occur in the 20s and that 20% of lesions are diagnosed before the age of 16, but I could not find this information in the literature, and the article did not provide an in-text citation for this.
In the case description (line 90):
3. You did not mention the ethnicity of the patient, since the AVMs you are discussing are most likely very common in a particular racial group, mentioning whether the patient belongs to this racial group (or not) would be of added value. (Kim H, Sidney S, McCulloch CE, Poon KY, Singh V, Johnston SC, Ko NU, Achrol AS, Lawton MT, Higashida RT, Young WL; UCSF BAVM Study Project. Racial/Ethnic differences in longitudinal risk of intracranial hemorrhage in brain arteriovenous malformation patients. Stroke. 2007 Sep;38(9):2430-7. doi: 10.1161/STROKEAHA.107.485573.)
4. You did not mention any information about the patient's family history. Since mounting evidence show that the disease is known to be hereditary. (Schimmel K, Ali MK, Tan SY, Teng J, Do HM, Steinberg GK, Stevenson DA, Spiekerkoetter E. Arteriovenous Malformations-Current Understanding of the Pathogenesis with Implications for Treatment. Int J Mol Sci. 2021 Aug 21;22(16):9037. doi: 10.3390/ijms22169037). Please indicate whether there is a history of this disease in the patient's family.
Minor comments:
Generally, the discussion section should be placed before the conclusion section.
Comments on the Quality of English LanguageGood
Author Response
The abstract needs to be slightly improved, with a clearer background narrative before the case presentation and a conclusion at the end.
Thank you for your feedback. We made some minor changes to the abstract to make it clearer for the reader.
1. Line 33, according to my knowledge AVMs occurs between arterioles and venules, without an intervening normal capillary network, but not between arteries and veins.
Thank you for your feedback. To our knowledge there is no textbook or literature available that states or specify that AVMs occour between arterioles and venules. All literature available to us states that AVMs occour between radiculomedullary arteries and veins without normal capillary network. Please, leave a report or a comment if we should change it.
2. Line 38-39, you mentioned that AVMs usually occur in the 20s and that 20% of lesions are diagnosed before the age of 16, but I could not find this information in the literature, and the article did not provide an in-text citation for this.
Thank you for your feedback. We missed a citation mark. Now it is added properly.
3. You did not mention the ethnicity of the patient, since the AVMs you are discussing are most likely very common in a particular racial group, mentioning whether the patient belongs to this racial group (or not) would be of added value. (Kim H, Sidney S, McCulloch CE, Poon KY, Singh V, Johnston SC, Ko NU, Achrol AS, Lawton MT, Higashida RT, Young WL; UCSF BAVM Study Project. Racial/Ethnic differences in longitudinal risk of intracranial hemorrhage in brain arteriovenous malformation patients. Stroke. 2007 Sep;38(9):2430-7. doi: 10.1161/STROKEAHA.107.485573.)
Thank you for highlighting this topic. We added ethnicity in our case report.
4. You did not mention any information about the patient's family history. Since mounting evidence show that the disease is known to be hereditary. (Schimmel K, Ali MK, Tan SY, Teng J, Do HM, Steinberg GK, Stevenson DA, Spiekerkoetter E. Arteriovenous Malformations-Current Understanding of the Pathogenesis with Implications for Treatment. Int J Mol Sci. 2021 Aug 21;22(16):9037. doi: 10.3390/ijms22169037). Please indicate whether there is a history of this disease in the patient's family.
Thank you for highlighting this topic. We added all the family history we have regarding the patient in our case report. Line 96-97.
Minor comments: Generally, the discussion section should be placed before the conclusion section.
We changed discussion section before the conclusion section.

Reviewer 2 Report
Comments and Suggestions for Authors
Dear authors,
The manuscript is interesting, but I do have some issues to address.
The introduction is very descriptive. Please, focus at the clinical case.
It is expected that the discussion contrasts the case with the current available literature. Please, enhance the section following the same parameters of a regular paper.
According to the WMA Declaration of Helsinki, which outlines ethical principles for medical research involving human subjects, informed consent is a critical requirement for research studies, including case reports.
Generally, a case report does not require Institutional Review Board (IRB) approval for publication because it is not considered “research” under the definitions used by the Department of Health and Human Services (DHHS). However, some journals may request a letter or acknowledgment from an IRB indicating whether IRB approval was obtained or not required for the described case. Please, make sure that Medicina is under such scope.
Author Response
1. The introduction is very descriptive. Please, focus at the clinical case.
Thank you for your feedback. Our focus in the introduction section is to highlight the importance of AVMs as well as to give basic information to the reader about the clinical case we are about to present in following sections. If necessary we can add short and summarized information about our clinical case in this section. Leave a report or a comment. Thank you.
2. It is expected that the discussion contrasts the case with the current available literature. Please, enhance the section following the same parameters of a regular paper.
Thank you for your feedback. We added more information in the discussion section with the current available literature regarding our case report.
3. According to the WMA Declaration of Helsinki, which outlines ethical principles for medical research involving human subjects, informed consent is a critical requirement for research studies, including case reports.
Thank you for your feedback. We have informed patient consent.
4. Generally, a case report does not require Institutional Review Board (IRB) approval for publication because it is not considered “research” under the definitions used by the Department of Health and Human Services (DHHS). However, some journals may request a letter or acknowledgment from an IRB indicating whether IRB approval was obtained or not required for the described case. Please, make sure that Medicina is under such scope.
To our knowledge Medicina journal does not require IRB approval for publication in a case report.

Reviewer 3 Report
Comments and Suggestions for Authors
This is a "standard" case presentation, with enough details and the course of the case well presented.
There are only to small observations that I make to the structure, they might be done in seconds, such as:
- for the figure 2, please switch the positions of the 2 images. A should, most logically, be placed at the left side of B
- please place the Conclusions section at the end of the article after the discussions and before the Author Contributions.
Overall, this paper is well done.
Author Response
- for the figure 2, please switch the positions of the 2 images. A should, most logically, be placed at the left side of B
Thank you for your feedback. We switched the positions of the images.
- please place the Conclusions section at the end of the article after the discussions and before the Author Contributions
Thank you for your feedback. We changed sections as mentioned above.
